# Relating parental stress with sleep disorders in parents and children

**Ray M. Merrill**[ID]*, **Kayla R. Slavik**

Department of Public Health, College of Life Sciences, Brigham Young University, Provo, Utah, United States of America

* Ray_Merrill@byu.edu

**Data Availability Statement:** All relevant data are within the paper and its Supporting Information files.

**Funding:** The authors received no specific funding for this work.

## Abstract

### Objective

To assess whether child sleep disorders positively correlate with parental insomnia, hypersomnia, and sleep apnea, and whether parental and child sleep disorders simultaneously positively associate with parental stress. Potential modifying influences of these associations by age, sex, and marital status will be considered.

### Methods

Analyses are based on 14,009 employees aged 18–64 with dependent children (n = 44,157) insured by Deseret Mutual Benefit Administrator (DMBA) in 2020. Rate ratios are adjusted for age, sex, and marital status.

### Results

The rate of parental stress is 3.00 (95% CI 2.33–4.85) times greater for those with insomnia and 1.88 (95% CI 1.59–2.22) times greater for those with sleep apnea. There is no increased risk of stress for those with hypersomnia. The number of dependent children filing one or more medical claims for a sleep disorder is 2.0%. Mean age is significantly older among those with a sleep disorder (17.1 vs. 14.4, t p < .0001). Child sex is not associated with the risk of having a sleep disorder. The rate of employee insomnia is 111% greater if their child has a sleep disorder, and employee sleep apnea is 115% greater if their child has a sleep disorder. The association between child sleep disorders and sleep apnea decreases with employee age (Wald chi-square p = 0.0410). The rate of employee stress is 90% greater if their child has a sleep disorder, 189% greater if they have insomnia, and 81% greater if they have sleep apnea. The strength of the association between insomnia and stress is greater for women (Wald Chi-square p = 0.0114), between sleep apnea and stress is greater for women (Wald chi-square p = 0.0010), and between sleep apnea and stress is greater for singles (Wald chi-square p = 0.0010).

### Conclusions

Better understanding the connection between parent and child sleep problems and parent stress, and modifying influences, may improve treatment of these disorders.

**Competing interests:** The authors have declared that no competing interests exist.

## Introduction

Chronic stress and adjustment disorders involving emotional or behavioral reactions to challenging life events can suppress the immune system by raising catecholamine and suppressor T cell levels and increase the risk of metabolic syndrome [1–8]. Immune response to stress can cause inflammation and contribute to other immune diseases such as psoriasis and rheumatoid arthritis [9]; trigger the release of histamine, which can cause severe broncho-constriction [10]; increase acid concentration in the stomach that contributes to ulcers [11]; and increase the risk of plaque buildup in the arteries, especially in individuals with a high-fat diet and sedentary lifestyle [12]. The link between chronic stress and psychological challenges is well established. Neurosis is a common mental health condition associated with chronic stress [1,13], as are anxiety, depression, OCD, bipolar disorder, and schizophrenia [1,13–20]. Stress also has been associated with sleep disorders [21].

Stress and sleep disorders are associated in two ways. First, stress promotes sleep disorders, and second, sleep disorders promote stress [22]. This correlation between stress and sleep can create a cycle that negatively affects an individual's health and quality of life. Stress has been associated with shortened or disturbed sleep, which can, consequently, create further symptoms of stress throughout the day [23]. Frequent occurrences of sleep disturbance can cause severe physical and psychological stress [24]. Studies have found a correlation between prolonged sleep latency, inability to stay asleep, and early morning awakening with various mental disorders such as depression [25]. Other studies also have found a connection between sleep apnea and depression [26,27].

Pediatric sleep disorders are more prevalent in children with underlying mental health conditions [28]. Childhood sleep disorders and mental health conditions, in turn, may contribute to increased parental stress [29–31]. Parents of children with sleep disorders report higher levels of daytime sleepiness (hypersomnia) and a negative impact on functioning [32]. Additionally, a greater association exists between maternal daytime sleepiness and child sleep than paternal daytime sleepiness [32]. Statistical modeling has shown that the quality of child sleep is a significant predictor of the quality of maternal sleep, as well as maternal mood, stress, and fatigue [33]. One study found that mothers with higher stress levels often took a longer time to fall asleep and had more perceived sleep problems. In turn, these mothers experienced more dysfunctional and less positive parenting [34]. Hence, lower levels of stress and corresponding improved sleep may promote more maternal involvement, responsiveness, and positive parenting.

Although studies have explored the association between child sleep disorders and parental hypersomnia and sleep quality in general, specific attention to insomnia and sleep apnea deserves consideration, along with potential modifying influences of these associations. In addition, while studies have shown a positive association between child sleep disorders and parental stress, they have not assessed whether child sleep disorders continue to correlate with parental stress after adjusting for parental sleep disorders. Potential modifying influences of these associations are also not well understood. The current study will explore the following hypotheses: 1. Child sleep disorders positively correlate with parental insomnia, hypersomnia, and sleep apnea; 2. These associations are modified by parental age, sex, and marital status; 3. Employee and child sleep disorder simultaneously positively correlate with parental stress; 4. These associations are modified by parental age, sex, and marital status.

## Material and methods

### Study population

The current study was based on employees and their dependent children receiving health insurance from Deseret Mutual Benefit Administrator (DMBA). The company was established

in 1970 to provide health insurance and retirement income to employees and their families of the Church of Jesus Christ of Latter-day Saints. In 2020, enrollees resided in Utah (72%), Idaho (9%), pacific states (10%), and other American states (9%). Job types for the employees included the Church education system, seminaries, and institutes (36%); manual labor (33%); other companies (11%); and other capacities (20%). A high percentage of employees were no longer insured through DMBA in the ages 65 or older as they became eligible for Medicare, a program under the U.S. Social Security Administration. Therefore, the current study is limited to adult employees under the age of 65 years.

## Data collection

The number of employees in the DMBA database aged 18 through 64 years was 21,027 in 2020. Approximately 23%, 42%, and 35% of employees in this age range had salaries less than $50 thousand, between $50 and $99 thousand, or at least $100,000, respectively.

The number of employees with dependent children (ages 0 through 25) was 14,009 (66.6%), with a total of 44,157 dependent children. Analyses were based on employees with dependent children. For these individuals, data were linked to automated healthcare claims from January 1 through December 31, 2020. The linked database was de-identified according to the Health Insurance Portability and Accountability Act (HIPAA) guidelines. Ethical approval and informed consent to participate was waived by the authors' institutional review board because the data were anonymized before assessment (IRB2021-157).

International Classification of Diseases, Tenth Revision, Clinical Modification (ICD-10-CM) codes were used to classify mental health conditions considered in this study [35]. Sleep disorders were identified in general (G47), along with specific types of sleep disorder: insomnia (G47.0), hypersomnia (G47.1), and sleep apnea (G47.3). Specific mental health disorders included stress (F43.0–F43.9), bipolar disorder (F31), depression (F32, F33), anxiety (F40, F41), obsessive-compulsive disorder (OCD) (F42), autism (F840), and attention deficit hyperactivity disorder (ADHD) (F90).

Rates of stress and specific types of mental illness and sleep disorders consisted of the number of enrollees filing one or more claims for each of the conditions divided by the number of enrollees. If multiple claims were filed in a given year for a specific condition, it was only counted once in the numerator of the rate calculation. Yet, an individual could contribute to more than one type of condition in a given year. Because only more serious levels of stress, mental illness, and sleep disorders tend to seek treatment, rates of these disorders will be lower than those provided by cross-sectional survey data.

Other variables considered in this study were employee age, sex, and marital status, and child age and sex. Salary was initially included but then dropped because it failed to significantly associate with employee stress or sleep disorders.

## Statistical techniques

Counts and percentages described the data. Rate ratios were calculated using Poisson regression. Rate ratios were calculated for employee stress and sleep disorders, adjusted for employee' age, sex, and marital status. Rate ratios also were calculated for child sleep disorders as the outcome variable, adjusted for child age and sex. Adjusted rate ratios were reported with corresponding 95% confidence intervals (CIs) in order to show the level of precision in the estimates and to assess statistical significance. The chi-square test, the Wald chi-square test, and the t-test were also used, with corresponding p-values reported. Two-sided tests of significance were used, based on the 0.05 level. Statistical analyses were derived from Statistical Analysis System (SAS) software, version 9.4 (SAS Institute Inc., Cary, NC, USA, 2012).

**Table 1. Employee rates of stress according to age, sex, and marital status, 2020.**

|  | Rate Ratio[a] | 95% LCL[a] | 95% UCL[a] |
|---|---|---|---|
| Child Sleep Disorder (Yes vs. No) | 1.90 | 1.33 | 2.72 |
| Insomnia (Yes vs. No) | 2.89 | 2.20 | 3.80 |
| Sleep Apnea (Yes vs. No) | 1.81 | 1.52 | 2.16 |
| Employee Age (Years) | 0.99 | 0.98 | 1.00 |
| Employee Sex (F vs. M) | 2.10 | 1.81 | 2.43 |
| Employee Marital Status (S vs. M) | 1.37 | 1.05 | 1.78 |
| Child Age (Years) | 1.00 | 0.99 | 1.02 |
| Child Sex (F vs. M) | 1.07 | 0.94 | 1.21 |

Data source: DMBA.

[a]Adjusted for the variables in the table. LCL: Lower Confidence Limit; UCL: Upper Confidence Limit. Hypersomnia is not included in the model because it did not significantly associate with child sleep or employee sleeping disorders.

## Results and discussion

The number of employees filing one or more medical claims for treating stress is 311 (2.2%). Women and singles experience higher rates of stress (Table 1). No significant interactions were identified in the model. Stress also is strongly associated with other more common mental health conditions like anxiety (1,251, 8.9%) and depression (1,129, 8.1%). Specifically, the rate of anxiety is 4.12 (95% CI 3.50–4.85) times greater in those with stress and the rate of depression is 4.00 (95% CI 3.46–4.78) times greater in those with stress.

The number of employees filing one or more medical claims for treating sleep disorders is 1,752 (12.5%): 301 (2.2%) for insomnia, 140 (1.0%) for hypersomnia, and 1,485 (10.6%) for sleep apnea. Older age is associated with higher rates of sleep disorders (Table 2). Men experience higher rates of sleep apnea. Singles have lower rates of sleep apnea, albeit statistically marginally insignificant. No significant interactions were identified in the models.

The rate of stress is 1.95 (95% CI 1.67–2.28) times greater in employees with a sleep disorder than in those without a sleep disorder, after adjusting for age, sex, and marital status. The

**Table 2. Employee rates of sleep disorders according to age, sex, and marital status, 2020.**

|  | Sleep disorders | | | Insomnia | | | Hypersomnia | | | Sleep Apnea | | |
|---|---|---|---|---|---|---|---|---|---|---|---|---|
|  | Rate Ratio[a] | 95% LCL[a] | 95% UCL[a] | Rate Ratio[a] | 95% LCL[a] | 95% UCL[a] | Rate Ratio[a] | 95% LCL[a] | 95% UCL[a] | Rate Ratio[a] | 95% LCL[a] | 95% UCL[a] |
| Age |  |  |  |  |  |  |  |  |  |  |  |  |
| 18–29 | 1.00 |  |  | 1.00 |  |  | 1.00 |  |  | 1.00 |  |  |
| 30–49 | 2.66 | 1.61 | 4.39 | 1.77 | 0.63 | 4.99 | 2.73 | 0.36 | 20.75 | 2.71 | 1.55 | 4.72 |
| 40–49 | 4.52 | 2.77 | 7.39 | 3.70 | 1.37 | 9.99 | 5.87 | 0.81 | 42.51 | 4.38 | 2.54 | 7.55 |
| 50–64 | 7.07 | 4.34 | 11.51 | 4.73 | 1.76 | 12.73 | 9.39 | 1.31 | 67.40 | 7.56 | 4.40 | 12.99 |
| Sex |  |  |  |  |  |  |  |  |  |  |  |  |
| Men | 1.00 |  |  | 1.00 |  |  | 1.00 |  |  | 1.00 |  |  |
| Women | 0.61 | 0.53 | 0.70 | 1.18 | 0.90 | 1.56 | 0.71 | 0.44 | 1.15 | 0.49 | 0.42 | 0.58 |
| Marital Status |  |  |  |  |  |  |  |  |  |  |  |  |
| Yes | 1.00 |  |  | 1.00 |  |  | 1.00 |  |  | 1.00 |  |  |
| No | 0.91 | 0.72 | 1.14 | 1.20 | 0.77 | 1.88 | 0.68 | 0.27 | 1.72 | 0.79 | 0.60 | 1.04 |

Data source: DMBA. LCL: Lower Confidence Limit; UCL: Upper Confidence Limit.

[a]Adjusted for employee age, sex, and marital status.

**Table 3. Child sleep disorders according to selected mental health conditions.**

| Child Mental Health Condition | No. | % | Sleep Disorder % | Rate Ratio[a] | 95% LCL[a] | 95% UCL[a] |
|---|---|---|---|---|---|---|
| Anxiety | 4567 | 10.34 | 6.53 | 6.05 | 5.20 | 7.04 |
| Depression | 3313 | 7.5 | 7.15 | 5.83 | 4.97 | 6.84 |
| ADHD | 1822 | 4.13 | 6.64 | 4.63 | 3.83 | 5.61 |
| OCD | 358 | 0.81 | 6.98 | 3.98 | 2.7 | 5.85 |
| Autism | 301 | 0.68 | 9.3 | 5.94 | 4.14 | 8.53 |
| Bipolar Disorder | 261 | 0.59 | 15.71 | 10.39 | 7.76 | 13.9 |
| Schizophrenia | 73 | 0.17 | 15.07 | 9.47 | 5.46 | 16.4 |
| Stress | 55 | 0.12 | 6.37 | 3.92 | 3.00 | 5.13 |

Data source: DMBA. LCL: Lower Confidence Limit; UCL: Upper Confidence Limit.

[a]Adjusted for the child's age and sex.

adjusted rate ratio for insomnia is 3.00 (95% CI 2.33–4.85), for hypersomnia is 0.83 (95% CI 0.42–1.65), and for sleep apnea is 1.88 (95% CI 1.59–2.22).

The number of dependent children filing one or more medical claims for a sleep disorder is 863 (2.0%). Mean age for dependent children is significantly older among those with a sleep disorder compared to without a sleep disorder (17.1 vs. 14.4, t p < .0001). Child sex is not associated with the risk of having a sleep disorder.

The rate of sleep disorders among children is significantly greater when the child has a mental health condition (Table 3). The strongest positive associations are in dependent children with bipolar disorder or schizophrenia. In addition, the rate of sleep disorders in the dependent children is 0.8% for those with no mental health conditions, 4.2% for those with one mental health condition, 6.8% for those with two mental health conditions, and 11.6% for those with three or more mental health conditions (Chi-square p < .0001).

Employee insomnia and sleep apnea are significantly associated with child sleep disorders (Table 4). The rate of employee insomnia is 111% greater if their child has a sleep disorder, after adjusting for the other variables shown in the table. The increased rate of employee sleep apnea is 115% greater if their child has a sleep disorder, after adjustment.

Interactions involving child sleep disorders and the other variables in the models were assessed. Only the association between child sleep disorders and sleep apnea was modified by

**Table 4. Rates of employee insomnia and sleep apnea positively associate with child sleep disorders.**

| | Sleep Disorder | | | Insomnia | | | Hypersomnia | | | Sleep Apnea | | |
|---|---|---|---|---|---|---|---|---|---|---|---|---|
| | Rate Ratio[a] | 95% LCL[a] | 95% UCL[a] | Rate Ratio[a] | 95% LCL[a] | 95% UCL[a] | Rate Ratio[a] | 95% LCL[a] | 95% UCL[a] | Rate Ratio[a] | 95% LCL[a] | 95% UCL[a] |
| Child Sleep Dis (Yes vs. No) | 2.25 | 1.89 | 2.69 | 2.11 | 1.47 | 3.04 | 1.13 | 0.58 | 2.21 | 2.15 | 1.78 | 2.60 |
| Employee Age (Years) | 1.05 | 1.04 | 1.05 | 1.04 | 1.03 | 1.05 | 1.05 | 1.04 | 1.07 | 1.05 | 1.04 | 1.06 |
| Employee Sex (F vs. M) | 0.58 | 0.53 | 0.64 | 1.31 | 1.10 | 1.56 | 0.81 | 0.61 | 1.09 | 0.46 | 0.41 | 0.52 |
| Employee Marital Status (S vs. M) | 0.79 | 0.66 | 0.95 | 1.18 | 0.87 | 1.61 | 0.40 | 0.19 | 0.86 | 0.65 | 0.52 | 0.81 |
| Child Age (Years) | 1.02 | 1.01 | 1.02 | 1.02 | 1.00 | 1.03 | 1.01 | 0.99 | 1.03 | 1.02 | 1.01 | 1.02 |
| Child Sex (F vs. M) | 1.01 | 0.95 | 1.07 | 0.97 | 0.85 | 1.11 | 1.08 | 0.90 | 1.31 | 1.01 | 0.95 | 1.07 |

Data source: DMBA. LCL: Lower Confidence Limit; UCL: Upper Confidence Limit.

[a]Adjusted for the variables in the table.

**Table 5. Employee rates of stress related to employee and child sleep disorders, 2020.**

|  | Rate Ratio[a] | 95% LCL[a] | 95% UCL[a] |
|---|---|---|---|
| Child Sleep Disorder (Yes vs. No) | 1.90 | 1.33 | 2.72 |
| Employee Insomnia (Yes vs. No) | 2.89 | 2.20 | 3.80 |
| Employee Sleep Apnea (Yes vs. No) | 1.81 | 1.52 | 2.16 |
| Employee Age (Years) | 0.99 | 0.98 | 1.00 |
| Employee Sex (F vs. M) | 2.10 | 1.81 | 2.43 |
| Employee Marital Status (S vs. M) | 1.37 | 1.05 | 1.78 |
| Child Age (Years) | 1.00 | 0.99 | 1.02 |
| Child Sex (F vs. M) | 1.07 | 0.94 | 1.21 |

Data source: DMBA. LCL: Lower Confidence Limit; UCL: Upper Confidence Limit. Hypersomnia is not included in the model because it did not significantly associate with child sleep or employee sleeping disorders.
[a]Adjusted for the variables in the table.

employee age, such that the significant positive association decreased with age (Wald chi-square p = 0.0410).

Employee stress is significantly associated with both employee and child sleep disorders (Table 5). The rate of employee stress is 90% greater if their child has a sleep disorder, 189% greater if they have insomnia, and 81% greater if they have sleep apnea, after adjusting for other variables shown in the table. A few significant interactions exist. Specifically, in the adjusted model the strength of the association between insomnia and stress is greater for women (Wald Chi-square p = 0.0114), the association between sleep apnea and stress is greater for women (Wald chi-square p = 0.0010), and the association between sleep apnea and stress is greater for singles (Wald chi-square p = 0.0010) (Fig 1).

Although some stress is a normal part of our daily function, continued chronic stress may cause or worsen mental health conditions such as anxiety and depression. Employees with stress were 312% more likely to experience anxiety and 300% more likely to experience depression. In the current study, medical claims related to stress involved 2.2% of the employees and sleep disorders involved 12.5% of employees. These are lower rates than would be identified through cross-sectional surveys because more severe cases tend to seek medical attention.

This study identified rates of parental stress and sleep disorders according to age, sex, and marital status. Higher rates of stress in women and singles is consistent with previous research [36]. We did not find that the association between sex and stress to be modified by marital status. Sleep disorders increased with age and men were more likely to have sleep apnea, as also consistent with previous research [37]. Higher levels of sleep apnea among married individuals approached statistical significance, which may be because having a sleeping partner increases the chance of having the disorder identified.

Stress was strongly positively associated with sleep disorders, as seen in other studies [21–27]. While the current study found significant positive associations between stress and insomnia and stress and sleep apnea, there was not a significant association between stress and hypersomnia. Other research has also found that insomnia is associated with stress [38], and that perceived stress is higher in patients with sleep apnea [39]. The null finding for hypersomnia is not supported by previous research.

Sleep disorders in children were positively associated with each of the mental health conditions, especially bipolar disorder and schizophrenia. The association between sleep disorders and mental health conditions in children has long been studied and recognized. Another recent study similarly reported that sleep disorders relate to all domains of mental health

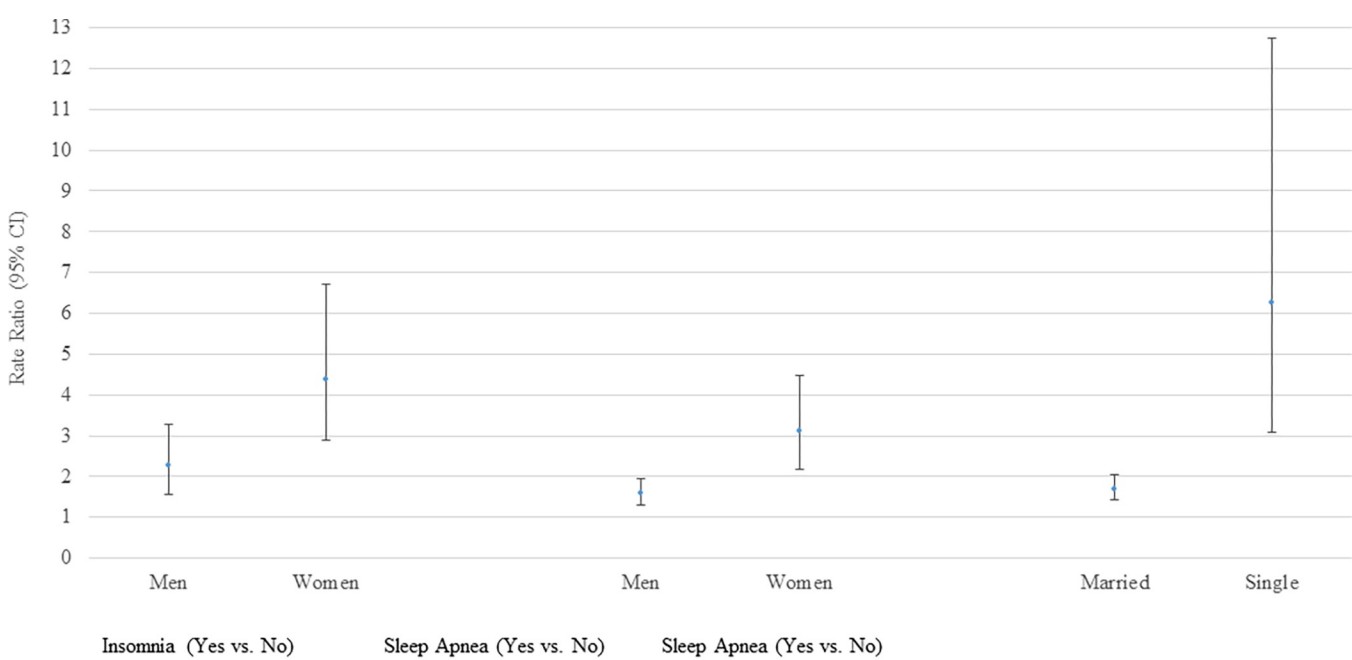

Adjusted for the variables in Table 4.

**Fig 1. Rates of employee stress by insomnia and sleep apnea modified by sex and marital status.**

conditions in children [40]. A review article identified a possible causal relationship between early childhood sleep disorders and mental health conditions, such as anxiety, depression, and ADHD in adolescents [41]. Sleep disturbance is a common symptom of bipolar disorder, such that during manic episodes there is a lower need for sleep and during periods of depression, insomnia may be experienced regularly [42,43]. In a prospective study, shorter total sleep time was associated with greater mania severity, and higher sleep variability was associated with increased mania and depression [44]. Another study found that disturbed sleep was a possible predisposing factor for bipolar disorder [45]. Profound disturbances in sleep continuity and architecture have been identified in patients with schizophrenia [46]. In a systematic review of the literature, one study found that the vast majority of patients with schizophrenia experience sleep disturbances, most commonly insomnia both prior to and over the course of the disease [47].

### Hypothesis 1. Child sleep disorders positively correlate with parental insomnia, hypersomnia, and sleep apnea

Child sleep disorders positively correlated with parental insomnia and sleep apnea, but not hypersomnia. When a child experiences a mental health condition or sleep disorder, the parent tends to be at greater risk of a sleep disorder. In general, parental stress arising from children with mental health or sleep problems is the likely mechanism influencing insomnia, hypersomnia, and sleep apnea. For example, having a child with a sleep disorder can be a stressful event. Other research has shown that parents of children with sleep problems experience higher levels of insomnia [48]. Mothers and fathers with insomnia report children with sleeping disorders such as bedtime resistance, sleep duration, sleep anxiety, waking at night, and/or daytime sleepiness [49]. As for the association between child mental health conditions and

parental sleep apnea, this association may be partly explained by the stress associated with the child mental health conditions making the symptoms of sleep apnea worse [50], thereby increasing the likelihood of a claim being filed for the sleep disorder. The null result involving hypersomnia is inconsistent with previous research [32,33] and deserves further investigation.

### Hypothesis 2. The associations identified in Hypothesis 1 are modified by parental age, sex, and marital status

Only employee age modified the association between child sleep disorders and employee sleep apnea. Inconsistent with our results, a previous study found that a positive relationship between child sleep disorders and hypersomnia exists in mothers but not fathers [32].

### Hypothesis 3. Employee and child sleep disorders simultaneously positively associate with parental stress

Child mental health conditions (including stress) increased the risk of child sleep disorders, and child sleep disorders was associated with significantly higher rates of parental stress, beyond that explained by employee sleep disorders. Child sleep disorders increased the rate of employee stress approximately 90%, while employee insomnia increased the rate 189% and employee sleep apnea 81%, after adjusting for selected demographic variables. Other research has found that parents who perceive their child as more demanding or more stressed have an increased risk of stress [26,27]. Psychological symptoms in children are often internalized by their parents, adding to parental stress and changing overall parental behavior [29]. A systematic review article involving children with ADHD or autism spectrum disorder found that child sleep problems were associated with higher parenting stress and poorer mental health [30].

### Hypothesis 4. The associations described in Hypothesis 3 are modified by parental age, sex, and marital status

Associations between child sleep disorders and employee stress were not significantly modified by employee age, sex, or marital status. However, associations between employee insomnia and stress and sleep apnea and stress were significantly higher for women than men, and the association between employee sleep apnea and stress was significantly higher for singles than married. These results may be because the severity of insomnia or sleep apnea is greater for women. Similarly, where it is more difficult to detect sleep apnea if there is not a sleep partner to help identify it, when it occurs in singles, it may be more serious, resulting in greater stress.

Potential limitations of the study warrant consideration. First, although bias due to under-reporting is unlikely because filing a claim will not result in loss of coverage, the full extent of stress and sleep disorders is not represented because only more serious cases are likely to seek medical care. Hence, generalization of the results should be restricted to severe stress and more serious sleep disorders. In addition, the study represents a single year and is therefore limited to identifying associations rather than cause-effect relationships. The bi-directional relationship between sleep disorders and stress is complicated and cannot be fully understood with a single year of data. However, future research may link additional years of data in order to better understand temporal sequences of events between sleep disorders and stress.

## Conclusions

The current study is based on medical claims data involving sleep disorders and stress. Thus, more severe cases are represented since these are the individuals more likely to seek medical

attention. Nevertheless, rates of stress and sleep disorders in parents varied according to age, sex, and marital status in a consistent manner to other studies. Results showing that stress strongly correlates with sleep disorders and that mental illness strongly correlates with sleep disorders in children also are consistent with previous research. Specific hypotheses tested in this study show that child sleep disorders positively correlate with parent insomnia and sleep apnea, but not hypersomnia; parent age modifies the association between child sleep disorders and parent sleep apnea; parent and child sleep disorders simultaneously correlate with parental stress; and the association between child sleep disorders and parental stress is not modified by parent age, sex, or marital status. In addition, associations between employee insomnia and stress and sleep apnea and stress are higher for women than men, and the association between employee sleep apnea and stress is higher for singles than married. Higher severity of insomnia or sleep apnea in women and in sleep apnea for singles may explain this latter result. Better understanding the connection between parent and child sleep quality and parent stress, and modifying influences, may help improve treatment and lower the risk of these disorders.

## Supporting information

**S1 File.**
(ZIP)

## Author Contributions

**Conceptualization:** Ray M. Merrill, Kayla R. Slavik.

**Data curation:** Ray M. Merrill.

**Formal analysis:** Ray M. Merrill.

**Writing – original draft:** Ray M. Merrill, Kayla R. Slavik.

**Writing – review & editing:** Ray M. Merrill, Kayla R. Slavik.

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
