## [Decision Letter · Decision Letter 0]

7 Sep 2022

PONE-D-22-15638Parental Stress and Sleep Disorders According to their Child’s Mental HealthPLOS ONE

Dear Dr. Merrill,

Thank you for submitting your manuscript to PLOS ONE. After careful consideration, we feel that it has merit but does not fully meet PLOS ONE’s publication criteria as it currently stands. Therefore, we invite you to submit a revised version of the manuscript that addresses the points raised during the review process.

Please note that we have only been able to secure a single reviewer to assess your manuscript. We are issuing a decision on your manuscript at this point to prevent further delays in the evaluation of your manuscript. Please be aware that the editor who handles your revised manuscript might find it necessary to invite additional reviewers to assess this work once the revised manuscript is submitted. However, we will aim to proceed on the basis of this single review if possible.

The reviewer has raised some concerns about the framing of the study's rationale and objectives, and has also suggested additional reporting of participant demographics. Please respond carefully to all of the points they have raised when preparing your revisions.

We look forward to receiving your revised manuscript.

Kind regards,

Jamie Males

Editorial Office

PLOS ONE

Journal Requirements:

3. Please ensure that you refer to Figure 1 in your text as, if accepted, production will need this reference to link the reader to the figure.

Reviewers' comments:

Reviewer's Responses to Questions

**Comments to the Author**

1. Is the manuscript technically sound, and do the data support the conclusions?

Reviewer #1: Yes

2. Has the statistical analysis been performed appropriately and rigorously? 

Reviewer #1: Yes

3. Have the authors made all data underlying the findings in their manuscript fully available?

Reviewer #1: Yes

4. Is the manuscript presented in an intelligible fashion and written in standard English?

Reviewer #1: Yes

5. Review Comments to the Author

Reviewer #1: This is an interesting subject and strong data collection, and I believe can be a good paper, yet, a major adjustment is needed before hand.

- I would think twice about the word “According” in the title. It directs readers in a different path.

- The objective is unclear.

- In the abstract there are 2 sentences that say the same thing –

o The rate of stress is 2.13 (95% CI 1.62–2.78) times higher in those with a sleep disorder after adjusting for age, sex, and marital status.

o Among contract holders, mothers, singles, and those with a sleep disorder had significantly higher rates of stress

If that is not so – than a clarification is needed.

- I suggest using the term “participants” instead of “contract holders”. This does not feel as an academic terminology.

- Conclusions -not innovative.

- Last sentence p2 – but also a possible predictor of forthcoming mental health disorders - why is this relevant? Needs clarifying.

- Hypotheses are needed.

- The introduction explains previous research on the given subject but does not show how this study adds on previous knowledge.

- Population –

o Data given is lacking. I don’t need to hear about the insurance company but data about the participants that may be relevant to stress and parenting. Such as: gender prevalence, mean age, socioeconomic status, marital status, family orientation, hours of work outside the house, etc..

- Discussion

This study identified the rate of parental stress and sleep disorders and the association between these variables. That is interesting but not adding to existing knowledge.

o Suggestions for future research are needed.

o What are the practical implications of this research?

6. PLOS authors have the option to publish the peer review history of their article (what does this mean?). If published, this will include your full peer review and any attached files.

Reviewer #1: No

---

## [Author Response · Author response to Decision Letter 0]

27 Oct 2022

Response to Reviewer Comments:

I would think twice about the word “According” in the title. It directs readers in a different path.

Response: The title has been changed to now say: “Relating Severe Stress with Sleep Disorders in Parents and Children”

- The objective is unclear

Response: The objective in the Abstract now says: “To assess whether child sleep disorder positively correlates with parental insomnia, hypersomnia, and sleep apnea, and to identify and whether parental and child sleep disorders simultaneously positively associate with parental stress. Potential modifying influences of these associations by age, sex, and marital status will be considered.” This objective is consistent with what is written at the end of the Introduction, in the Discussion, and in the Conclusion.

- In the abstract there are 2 sentences that say the same thing –

The rate of stress is 2.13 (95% CI 1.62–2.78) times higher in those with a sleep disorder after adjusting for age, sex, and marital status.

Response: The Abstract has been rewritten to better align with the clarified objective and hypotheses of the study. These sentences are no longer included.

Among contract holders, mothers, singles, and those with a sleep disorder had significantly higher rates of stress

If that is not so – then a clarification is needed.

Response: This statement is consistent with the findings.

- I suggest using the term “participants” instead of “contract holders”. This does not feel as an academic terminology.

Response: Because all enrollees (parents and children) in the study are participants, and because we wanted to distinguish between the contract holders and the children, we thought the word “employees” would work better for “contract holders.”

- Conclusions -not innovative.

Response: the Conclusion section in the Abstract and at the end of the paper tries to emphasize the importance of this study.

- Last sentence p2 – but also a possible predictor of forthcoming mental health disorders - why is this relevant? Needs clarifying.

Response: The sentence was deleted.

- Hypotheses are needed.

Response: Four specific hypotheses were added at the end of the Introduction. The Discussion is organized by these four hypotheses.

- The introduction explains previous research on the given subject but does not show how this study adds on previous knowledge.

Response: The last paragraph of the Introduction has been rewritten and clarifies what this study adds.

- Population –

o Data given is lacking. I don’t need to hear about the insurance company but data about the participants that may be relevant to stress and parenting. Such as: gender prevalence, mean age, socioeconomic status, marital status, family orientation, hours of work outside the house, etc.

Response: Under the section titled Data Collection in the Methods, a little more information about the study group is added (i.e., the distribution of salaries for the employees). The first paragraph of the Methods includes a description of the enrollees in DMA according to place of residence and job type. 

- Discussion

This study identified the rate of parental stress and sleep disorders and the association between these variables. That is interesting but not adding to existing knowledge.

Response: The Introduction (last paragraph) now clarifies what is known and what will be added. The Discussion begins by addressing how the Results provide information consistent with the literature. Discussion about the hypotheses, however, provides information on what knowledge is being added by this study.

o Suggestions for future research are needed.

Response: In the Discussion we add that “The null result involving hypersomnia is inconsistent with previous research [32, 33] and deserves further investigation.” Also, a limitation paragraph was added at the end of the Discussion. One sentence included there says: “However, future research may link additional years of data in order to better understand temporal sequences of events between sleep disorders and stress.”

o What are the practical implications of this research?

Response: The following was added at the end of the Conclusion: “Better understanding the connection between parent and child sleep quality and parent stress, and modifying influences, may help improve treatment and lower the risk of these disorders.”

---

## [Decision Letter · Decision Letter 1]

8 Dec 2022

Relating Severe Stress with Sleep Disorders in Parents and Children

PONE-D-22-15638R1

Dear Dr. Merrill,

We’re pleased to inform you that your manuscript has been judged scientifically suitable for publication and will be formally accepted for publication once it meets all outstanding technical requirements.

Kind regards,

Gudmundur Skarphedinsson, PhD

Academic Editor

PLOS ONE

Additional Editor Comments (optional):

Reviewers' comments:

Reviewer's Responses to Questions

**Comments to the Author**

1. If the authors have adequately addressed your comments raised in a previous round of review and you feel that this manuscript is now acceptable for publication, you may indicate that here to bypass the “Comments to the Author” section, enter your conflict of interest statement in the “Confidential to Editor” section, and submit your "Accept" recommendation.

Reviewer #1: All comments have been addressed

2. Is the manuscript technically sound, and do the data support the conclusions?

Reviewer #1: Yes

3. Has the statistical analysis been performed appropriately and rigorously? 

Reviewer #1: Yes

4. Have the authors made all data underlying the findings in their manuscript fully available?

Reviewer #1: Yes

5. Is the manuscript presented in an intelligible fashion and written in standard English?

Reviewer #1: Yes

6. Review Comments to the Author

Reviewer #1: I have found that all of my concerns have been addressed. The authors have added research questions and have cleared their intent as well as the implications of this study.

7. PLOS authors have the option to publish the peer review history of their article (what does this mean?). If published, this will include your full peer review and any attached files.

Reviewer #1: No

---

## [Editor Report · Acceptance letter]

19 Dec 2022

PONE-D-22-15638R1 

Relating Parental Stress with Sleep Disorders in Parents and Children 

Dear Dr. Merrill:

I'm pleased to inform you that your manuscript has been deemed suitable for publication in PLOS ONE. Congratulations! Your manuscript is now with our production department. 

Kind regards, 

on behalf of

Dr. Gudmundur Skarphedinsson 

Academic Editor

PLOS ONE